
# Global partitioning of runoff generation mechanisms using remote sensing data

Joseph T.D. Lucey[1,2], John T. Reager[2], Sonya R. Lopez [1]

[1]Department of Civil Engineering, California State University, Los Angeles, Los Angeles, California, 90032, USA
[2]NASA Jet Propulsion Laboratory, California Institute of Technology, Pasadena, California, 91109, USA

Correspondence to: J.T. Reager (John.Reager@jpl.nasa.gov)

**Abstract.** A set of complex processes contribute to generate river runoff, which in the hydrological sciences are typically divided into two major categories: surface runoff, sometimes called Hortonian flow, and baseflow-driven runoff or Dunne flow. In this study, we examine the covariance of global satellite-based surface water inundation observations with two
remotely sensed hydrological variables, precipitation, and terrestrial water storage, to better understand how apparent runoff generation responds to these two dominant forcing mechanisms. Terrestrial water storage observations come from NASA's GRACE mission, while precipitation comes from the GPCP combined product, and surface inundation levels from the NASA SWAMPS product. We evaluate the statistical relationship between surface water inundation, total water storage anomalies, and precipitation values under different time lag and quality control adjustments between the data products. We find that the
global prediction of surface inundation improves when considering a quality control threshold of 50% reliability for the SWAMPS data, and after applying time lags ranging from 1 to 5 months. Precipitation tends to be the dominant driver of surface water formation at zero time lag in most locations, while very wet tropical locations and high latitudes also contain a storage driven runoff component at variable time lags.

## 1 Introduction

There is a long history of research concerning the mechanisms that control runoff generation at the terrestrial land surface (e.g. Beven and Kirkby, 1976; Pearce et al., 1986; Lyon et al., 2006; Vivoni et al., 2007; Kirchner, 2009). In brief, it is generally well accepted that two major mechanisms are responsible for surface water formation: (1) excess precipitation and the limitation of infiltration causing surface runoff, or (2) the rising of the water table and deeper soil moisture to push more water into stream networks at low topography. If precipitation rates exceed infiltration rates, then precipitation dominates surface
inundation development and is typically defined as Hortonian flow. If precipitation successfully infiltrates and soils become saturated, then subsurface soil water storage will dominate surface water formation, typically described as Dunne flow. These are core concepts within terrestrial hydrology; however, there are limited observational studies on these runoff generation mechanisms at scales larger than a catchment. We are not aware of any studies that have assessed the contributions to surface water formation over a global domain. However, using existing data on global precipitation and water storage, and considering





how these two mechanisms influence surface inundation development, it is now possible to examine surface runoff mechanisms across a range of land surface conditions.

Satellite observations offer a means to observe changes in hydrology over a global domain, presenting a distinct advantage over in-situ observations in representing a variety of hydrological mechanisms and processes across ecosystems and land cover types. Previously published work has utilized a variety of measurements of catchment or basin antecedent

conditions, such as soil moisture or vertically integrated water storage, to assess the influence of soil water on runoff generation (e.g. Koster et al., 2010; Reager et al., 2014). NASA's Gravity Recovery and Climate Experiment (GRACE) mission (Tapley et al., 2004) offers a 15+ year observational record on the state of terrestrial water storage globally. GRACE measures a change in the gravitational potential that is often linearly related to the amount of water stored at the land surface beneath the satellites. While these measurements are increasingly uncertain at resolutions beneath ~150,000 $km^2$, they offer a robust and highly

accurate means to measure changes in storage for areas larger than 150,000 $km^2$ (e.g. Wahr et al., 2006; Wiese et al., 2016) and offer a globally gridded data set of terrestrial water storage anomalies (TWSA) that is relatively easy to use. Previously, GRACE observations have been applied to develop a flood potential index and to characterize the intensity of certain flood events based on storage pre-conditioning or "flood potential" (Reager et al., 2009; Reager et al., 2014). These studies serve as proof that integrated basin water storage is significant in understanding surface inundation changes.

There is also extensive literature relating to the influence of precipitation on surface inundation (Guo et al., 2012; Kirchner, 2009). The Global Precipitation Climatology Project (GPCP) offers a globally gridded precipitation dataset that optimally combines satellite, in situ and land radar measurements into a single best product (Adler et al., 2003). This precipitation data set can be used to assess the relationship between rainfall and surface water inundation globally.

The satellite observations of TWSA and precipitation can be related to observations of surface water formation from

the Surface WAter Microwave Product Series (SWAMPS) (Schroeder et al., 2014) dataset to better understand runoff generation. SWAMPS was created based on optical and radiometric observations of surface reflectance that are often associated with water. These observations are expressed in terms of fractional inundation, or the percentage of land occupied by surface water at a 0.25-degree grid resolution globally. Schroeder et al. (2014) provide a quality control map expressed as likelihood or confidence that allows a user to mask out unreliable data at the quality threshold of their choosing.

There are no previous studies on the hypothesized linear relationships between precipitation, storage and surface inundation across the globe. We conduct such a study here too: (1) assess the viability of satellite data to quantify this relationship; (2) determine which mechanism has the more considerable influence in different regions, (3) characterize general behavior. We approach these goals through the development of a simple linear model of inundation based on remote sensing observations.





## 2 Data and Methods


The datasets downloaded for this work include surface inundation (Surface WAter Microwave Product Series; SWAMPS), global precipitation estimates (Global Precipitation and Climatology Project; GPCP), and groundwater storage (Gravity Recovery and Climate Experiment; GRACE).

SWAMPS is available from Columbia University at approximately 0.25° x 0.25° [approx. 25 km x 25 km] spatial

resolution and daily temporal resolution from February 1st, 1992 to January 31st, 2017. The SWAMPS dataset reports a quality control map that represents the reliability of their published fractional surface water, which is influential in our reported results (Schroeder et al., 2014) (Fig. 1a). Desert land covers have low reliability in their inundation measurements. The Sahara Desert has explicitly poor measurements due to limestone deposits. Other variables that were reported to interfere with the SWAMPS signal were snow and precipitating clouds.

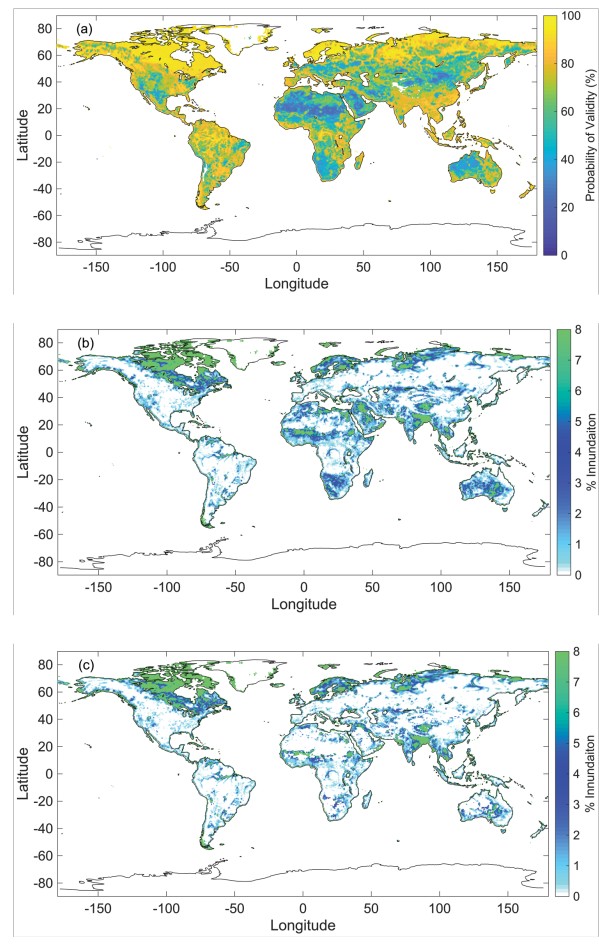


**Figure 1: a) SWAMPS quality control map. b) Example of monthly SWAMPS measurements for August 2007. c) Fig. 1b after locations less than 50% probability of validity are removed.**





GPCP is available from the National Oceanic & Atmospheric Administration's (NOAA) Earth System Research Laboratory at 2.5° x 2.5° [approx. 250 km x 250 km] spatial resolution and monthly temporal resolution from January 1979 to present (Adler et al., 2003). GPCP provides global precipitation measurements in mm/day (Fig. 2a).

GRACE measures the gravity anomaly detected by the orbiting satellites; the JPL GRACE Tellus group processes the anomalies and provides the change in total water storage across the globe [cm] (Fig. 2b). GRACE is available at a 3.0° x 3.0° [approx. 300 km x 300 km] spatial resolution and monthly temporal resolution from April 2002 to June 2017 (Watkins et
al., 2015, Wiese et al., 2016).

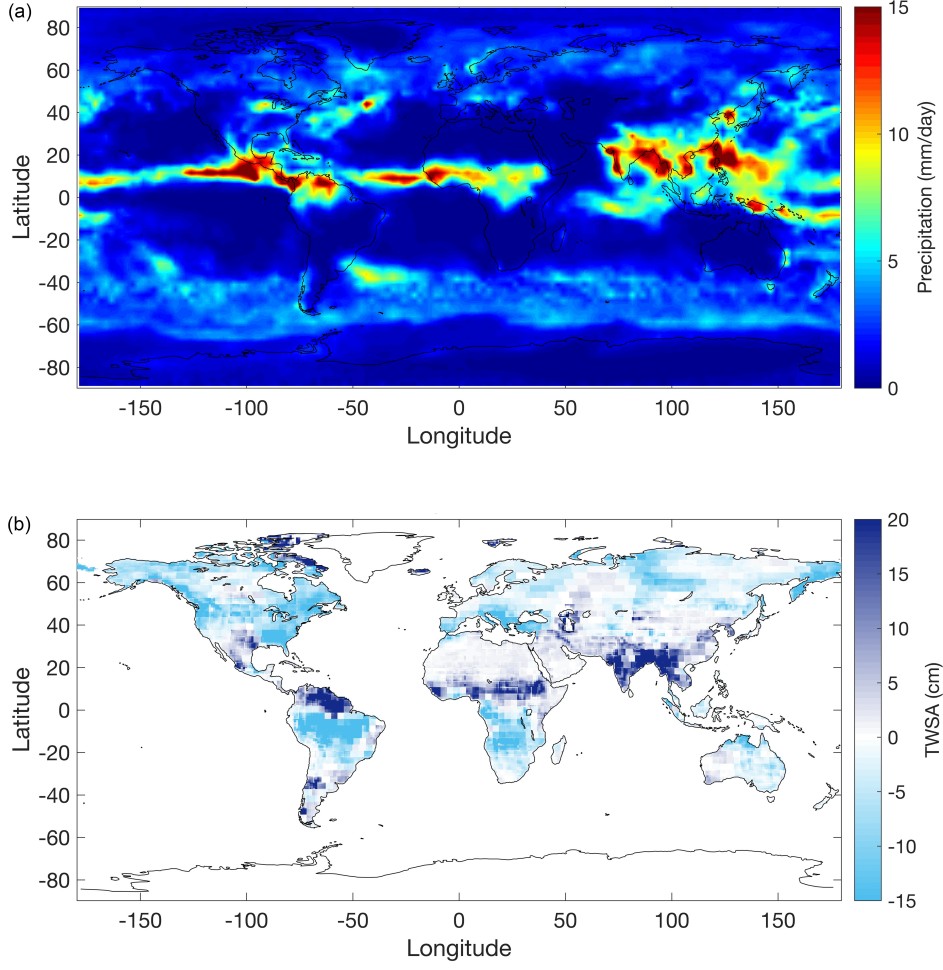

**Figure 2: a) Example of monthly GPCP measurements for August 2007. b) Example of monthly GRACE total water storage anomaly (TWSA) measurements for August 2007.**





85       After data acquisition, our preliminary step was to re-grid each dataset using linear interpolation to a common 0.5° x 0.5° spatial resolution. Also, we averaged daily surface inundation measurements from SWAMPS to achieve monthly values. The timeframe for this work spanned April 2002 to October 2015, the common period amongst these products.

This work involved assessing the viability of a single-linear regression (Eq. (1) and (2)), or multi-linear regression (Eq. (3)) model based on GPCP and GRACE, to predict surface inundation estimated by SWAMPS.

$$SWAMPS = m(GPCP) + b \qquad (1)$$

$$SWAMPS = m(GRACE) + b \qquad (2)$$

$$SWAMPS = m_1(GPCP) + m_2(GRACE) + b \qquad (3)$$

      Using the correlation coefficients ($R^2$) and regression coefficients (slope values; $m$, $m_1$, and $m_2$), we can statistically determine which mechanism will have a stronger influence on surface inundation developments. To further develop a model

capable of capturing long-term variability across the globe, we utilized each dataset's climatology.

      To develop these climatology datasets, we calculate the long-term monthly average values. The resulting dataset would be a single value at each cell for each month, reflecting the average monthly signal occurring through the historical record. Using the climatology, we can observe the average annual hydrologic cycle anywhere across the globe.

      After completing the regressions, multiple grid cells had negative regression coefficients. Negative regression

coefficients are of concern because it should generally be impossible to have an inverse relationship between surface inundation and precipitation or groundwater storage. In most cases, time-lags between forcing and response (for example a high TWSA due to snow which only manifests as surface water 3 months later) are responsible for negative regression coefficients within the developed model and applying optimal lag corrected correlations improved our statistical strengths. We conducted iterative cross-correlations between TWSA and inundation and between precipitation and inundation to statistically determine the most

appropriate time correction at each cell location across the globe (Fig. 4). We applied two time-lag thresholds: 0 to 5 months and 0 to 11 months lag. Time lag corrections occur at each grid cell, which shifts the climatology signal of GRACE or GPCP within the phase of SWAMPS.

      The final step in pre-processing the datasets is the removal of low-quality data from the SWAMPS dataset. Schroeder et al. (2014), issued a quality control (QC) map for the SWAMPS dataset (Fig. 1a) and this we set the quality threshold at 50%

confidence or higher. As previously stated, desert regions (i.e., Sahara Desert, Southern Africa, and Western Australia) and snow-dominated regions (i.e., Rocky Mountains and Central Asia) have poor reliability in measurements, likely due to erroneous reflectivity, and are largely filtered out from the study domain (Fig. 1b and 1c).

      In total, nine regression models were validated by calculating surface inundation and comparing to the SWAMPS dataset. Pearson's $R^2$, the root mean squared error (RMSE), coverage and a ratio between $R^2$ and coverage were used to

determine each model's strength. We determined coverage by counting the number of cells within the global polygon; this analysis excluded Antarctica and Greenland which had no SWAMPS coverage. A model with a ratio closer to one describes a stronger model; this ratio is important because it considers maximizing coverage and correlation to observations. In choosing





the 'best' model, we are considering two things: (1) overall model performance at predicting surface inundation, and (2) the global coverage retained. With the final model, historical GRACE and GPCP measurements are used to calculate modeled surface inundation. A best-fit line is applied to display the relationship between modeled surface inundation and measured SWAMPS values.

After selecting the best model, we assessed model performance on a basin and global scale. Correlation statistics ($R^2$ and RMSE) between measured and model climatologies and scatterplots are used to present model performance at four highly studied basins: Amazon River in South America, Mackenzie River in Canada, Mississippi River in the USA, and Ob River in Russia. The difference between modeled and measured surface inundation highlights locations of over and under predictions across the global domain.We estimated the root-mean-squared error (RMSE) between modeled and measured surface inundation for our entire observational period to evaluate our model's error in predictions across the historical record. Finally, the relative error of SWAMPS was calculated using Eq. (4) to determine the error between modeled and measured SWAMPS relative to the measured SWAMPS signal.

We took the difference between normalized GPCP and GRACE slopes to determine whether groundwater storage or precipitation is relatively more influential in surface inundation developments. These variables were standardized to compare them on the same scale (Eq. (5)). Equation (6) is used to compare the standardized slopes. Flows were classified as Horton flows if the value was positive (i.e. precipitation was dominant in runoff generation). Flows were classified as Dunne flows if the value was negative (i.e. TWSA was dominant in runoff generation). Values closer to zero will show that both groundwater storage and precipitation are both equally important in surface inundation developments at that location. The methodology is displayed as a flowchart in Figure 3 to clarify our process further.

$$Error\ (\%) = \frac{RMSE}{LTA} \tag{4}$$

$$Standardized\ Values = \frac{x - \mu}{\sigma} \tag{5}$$

$$Control\ Variable = |GPCP| - |GRACE| \tag{6}$$





```
                    ┌─────────────────┐
                    │ Data acquisition:│
                    │ SWAMPS, GPCP,   │
                    │ GRACE           │
                    └─────────────────┘
                            │
                    ┌─────────────────┐
                    │ Create monthly  │
                    │ average values for│
                    │ SWAMPS          │
                    └─────────────────┘
                            │
                    ┌─────────────────┐
                    │ Interpolate to  │
                    │ 0.5° x 0.5° grid│
                    └─────────────────┘
                            │
                    ┌─────────────────┐
                    │ Create climatology│
                    │ datasets        │
                    └─────────────────┘
```

Data acquisition: SWAMPS, GPCP, GRACE

Create monthly average values for SWAMPS

Interpolate to 0.5° x 0.5° grid

Create climatology datasets

Time lag corrections (Cross-correlations)

Single-linear regression

Multi-linear regression

Calculate inundation with developed models

Quality Control

Model performance metrics

Additional model performance with selected model

Determine control variable

Figure 3: Methodology flowchart.



## 3 RESULTS

Lag maps display the signal lag between SWAMPS and GRACE or SWAMPS and GPCP for 0 to 11 months (Fig. 4a and 4b)
and 0 to 5 months (Fig. 4c and 4d). Locations in the white represent no lag or no data and areas in red represent long delays.
The color-axis range is from 0 to 5 months of lag.  We can see minimal differences comparing the lags maps for 0 to 11 months
correction and 0 to 5 months correction. Majority of the GRACE and GPCP signal is only out of phase with SWAMPS by at
most five months. This is statistically supported in Table 1 because $R^2$ and RMSE from all 0 to 11 month scenarios match their
0 to 5 month time lag counterpart. We no longer considered all 0 to 11 month models beyond this point.

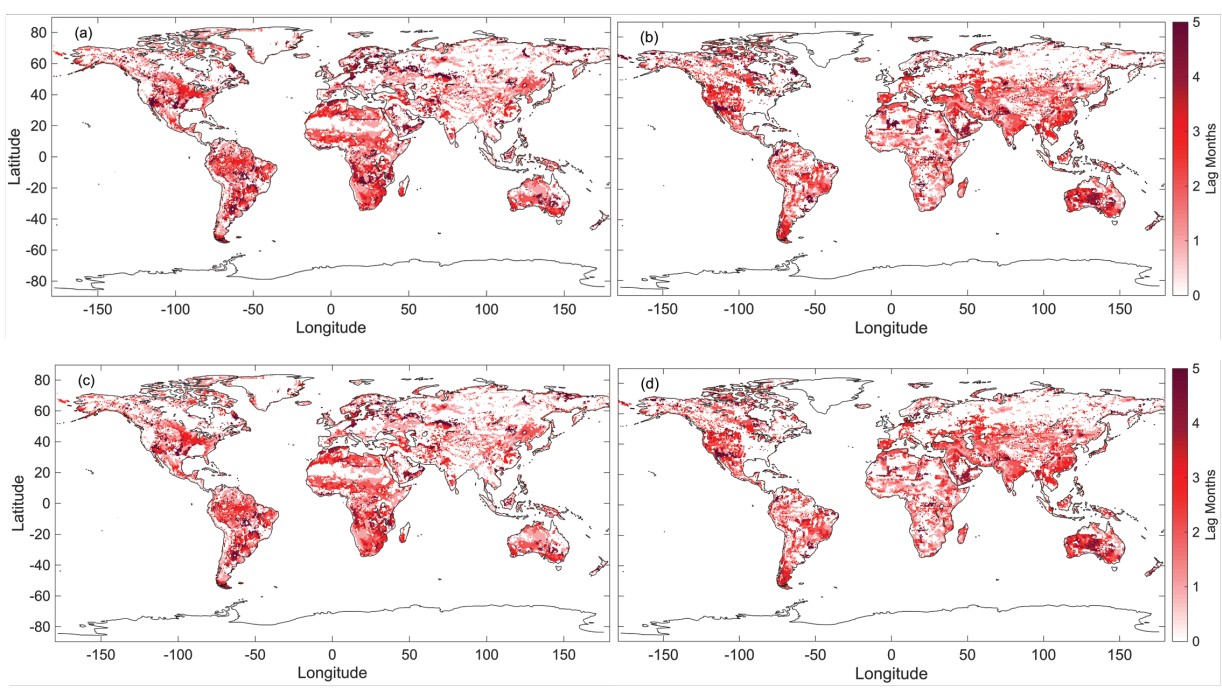

**Figure 4: Maps display the number of months between SWAMPS, GRACE, and GPCP signal that were statistically
determined by cross-correlations. a) GPCP lag map with a time threshold of 0 to 11 months. b) GRACE lag
map with a time threshold of 0 to 11 months. c) GPCP lag map with a time threshold of 0 to 5 months. d)**
**GRACE lag map with a time threshold of 0 to 5 months.**

Measured and modeled SWAMPS values are displayed using scatterplots (Fig. 5). The x-axis displays modeled
SWAMPS while the y-axis represents measured SWAMPS. These plots reveal global surface inundation measurements from
April 2002 to October 2015 without the consideration of quality control, referred to as QC, (Fig. 5a) and with QC (Fig. 5b).
The red line displays the best fit relationship as determined by MATLAB's statistical toolbox. We can statistically and visually
see the significance of removing locations with less than 50% QC. The $R^2$ increased (0.732 to 0.900) and RMSE decreased



(3.830 to 1.890) after QC was applied (Fig. 5). There is a large spread of surface inundation from the model (Fig. 5a), but after masking there is a clear trend line between modeled and measured SWAMPS (Fig. 5b). Further comparing the validation statistics between single and multi-linear models, we can see there isn't much improvement. However, we know that a model
with both GRACE and GPCP better represents the world compared to just considering one variable. A multi-linear regression model with a time lag correction improves in both RMSE and R² compared to the non-time corrected. Therefore, a multi-linear regression model with a time lag correction between 0 to 5 months is the most rigorous model for further analysis.

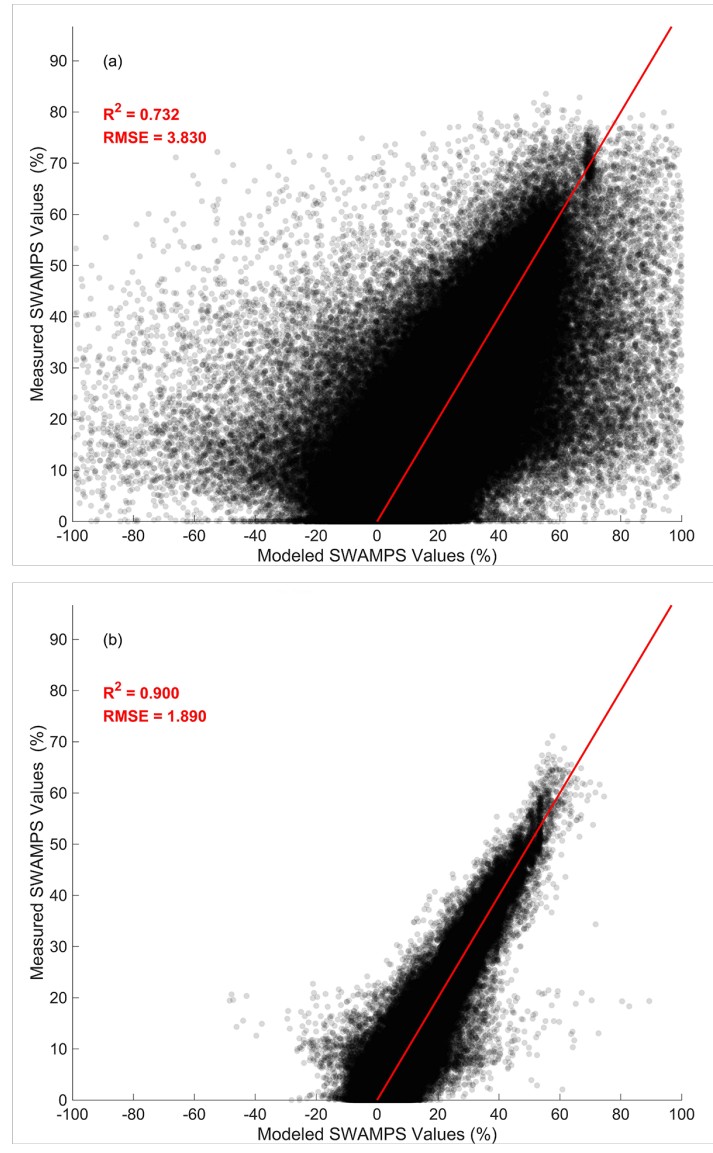

**Figure 5: Example of multi-linear regression model validation plots. a) Measured versus modeled SWAMPS with a**
**time lag correction of 0 to 5 months b) Fig. 5a after locations less than 50% probability of validity are removed.**





Modeled SWAMPS using GRACE and GPCP (Fig. 6a) and measured SWAMPS (Fig. 6b) are displayed with a time lag correction between 0 and 5 months during August 15th, 2007. Green locations are reported to have high inundation values while white spots have low inundation values or no available data. The percent difference between these two maps (Fig. 6c)

identifies locations of over and underestimation. The red, grey, and blue locations represent overestimations, minimal differences, and underestimations, respectively, between modeled and measured inundation. Majority of the domain is grey because the differences between small values of inundation are insignificant. Modeled SWAMPS has the largest limitations at locations with snow or ice (around the Great Lakes and northern parts of Russia) and in areas that experience seasonal monsoons (Bay of Bengal and west coast of South Africa).

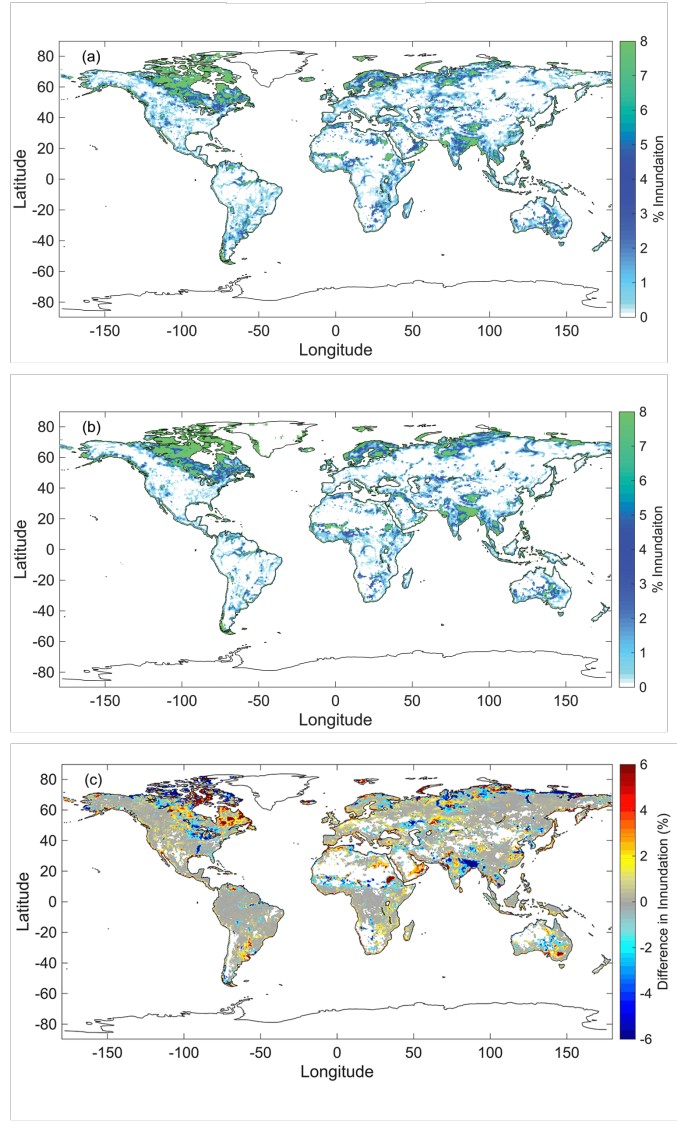




**Figure 6: Visual comparison of monthly modeled and measured SWAMPS. a) Modeled surface inundation. b) Measured surface inundation. c) The absolute difference between modeled and measured surface inundation. Modeled SWAMPS has a time correction of 0 to 5 months.**

185        Regional model performance is assessed through correlation statistics between climatologies and scatterplots for measured and modeled inundation (Fig. 7). The Amazon (Fig. 7a-c), Mackenzie (Fig. 7d-f), Mississippi (Fig. 7g-7i), and Ob (Fig. 7j-l) River Basins were used for this analysis because their hydrology is well understood and a successful model should maintain its rigor in these significant areas. Blue, red, and green markers (Fig. 7a, 7d, 7g, and 7j) represent randomly selected cell locations along the river, measured and modeled climatologies are represented with solid and dashed lines using the same

color scheme (Fig. 7b, 7e, 7h, and 7k); the cell coordinates are in Table 2. Red boxes (Fig. 7a, 7d, 7g, and 7j) outline the cells used in the scatterplots (Fig. 7c, 7f, 7i, and 7l) and their boundary coordinates are also in Table 2. Climatology correlation statistics are in Table 3. Similar to Figure 5b, the scatterplots relate measured and modeled inundation between April 2002 to October 2015 with QC applied for the cells within the boundaries. The red line displays the best fit line along with the calculated R2. The multi-linear regression model with a time lag correction between 0 to 5 months is used to calculate modeled

inundation. Majority of the basins' domains display strong statistics between the measured and modeled inundation (Table 3). Basins that experience varying snow seasons (Mississippi and Ob) have the largest modeled and measured inundation discrepancies (Fig. 7i and 7l). These two river basins have the largest spread in modelled versus measured about the best fit line and have reduced R2 correlations (0.511 and 0.629, respectively). Inadequate data during the snow season is limiting model performance during these times (no available measurements during winter months as seen in Fig. 7e and 7k).



**Figure 7: Cells included in scatter plots are outlined by the red boxes and red, blue, and green dots denote the cell used for measured and modeled climatologies. Modeled inundation has a time correction of 0 to 5 months. a) Amazon map. b) Amazon measured and modeled climatologies. c) Amazon scatterplot. d) Mackenzie map. e) Mackenzie measured and modeled climatologies. f) Mackenzie scatterplot. g) Mississippi map. h) Mississippi measured and modeled climatologies. i) Mississippi scatterplot. j) Ob map. k) Ob measured and modeled climatologies. l) Ob scatterplot.**







To assess global model performance, we calculate the RMSE (Fig. 8a) between the measured and modeled time series at each grid cell. Low RMSE values represent small differences between long-term modeled and measure SWAMPS while high RMSE values tell us there are more considerable differences in the signals. Grey represents low error values while red displays more substantial error. White locations have no value. Long-term surface inundation (Fig. 8b) values range from 0 to 8% with high values in green, low values and no value in white. Figure 8c displays errors in our modeled SWAMPS relative

to the measured SWAMPS signal. Locations with heavy snow (northern parts of North America, Europe, and Central Asia) and regular annual cycles of inundation (India and Amazon) have more significant RMSE values compared to other locations.

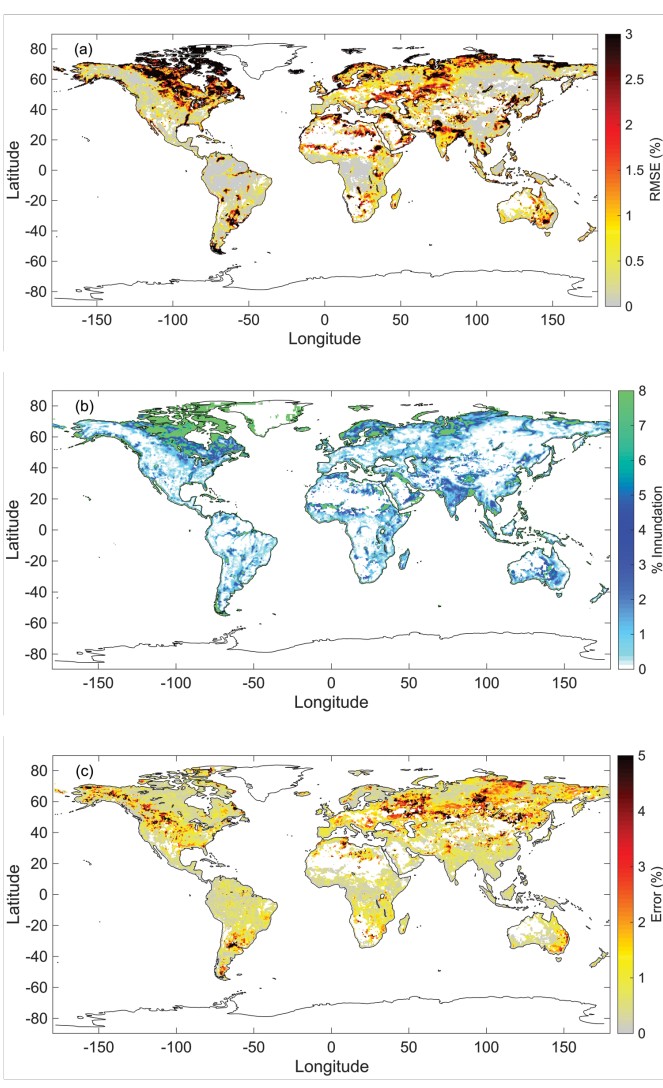



**Figure 8: a) RMSE between modeled and measured SWAMPS with time correction of 0 to 5 months. b) Long-term average (LTA) surface inundation. c) Error relative to the measured SWAMPS signal.**


Depending on the global location, either GRACE, GPCP or both control surface inundation for the no time-lag correction (Fig. 9a), 0 to 5 months (Fig. 9b), and 0 to 11 month corrected models (Fig. 9c). Precipitation dominate locations are red, and groundwater storage controls blue locations. Grey areas represent locations controlled by both GRACE and GPCP. Areas shown in white represent no values. Overall, we determined that both GPCP and GRACE control majority of surface

inundation developments across the world. By taking the standard deviation ($\sigma$) of the standardized modeled SWAMPS values ($\sigma = 1.04$), we determined the percentage of cells controlled by GRACE, GPCP or both. Cells with a difference less than our calculated standard deviation ($-\sigma$) were considered GRACE dominate. Cells with a difference greater than our calculated standard deviation ($+\sigma$) were GPCP dominate. Both groundwater and precipitation controlled cells have values within $\pm\sigma$. Using these standards, we found groundwater storage controlled 8.3% of cells which produced Dunne flows. Precipitation

controlled 6.9% of cells and generated Horton flows. Both variables controlled approximately 84.8% of cells.





**Figure 9: Control variable maps with a) no time correction, b) time correction of 0 to 5 months, and c) time correction 0 to 11 months.**





Maps with correlation values (Fig. 10, 11a, and 11b) have a color-axis from 0 to 1. Correlations closer to 1, displayed in yellow, represent stronger relationships between SWAMPS and the other dataset(s). Correlations closer to 0, presented in blue, represent weaker relationships between SWAMPS and the other datasets(s). We provided five correlation maps with different inputs: the no time-lag corrected model with SWAMPS and GRACE (Fig. 10a), the no time-lag corrected model with SWAMPS and GPCP (Fig. 10b), the no time-lag corrected model with SWAMPS, GRACE and GPCP (Fig. 10c and 11a), and
the 0 to 5 month time corrected model with SWAMPS, GRACE, and GPCP (Fig. 11b).

Correlation maps from the single linear regressions comparing (Fig. 10a, and 10b), demonstrate limitations in correlation strengths. Using GRACE alone, there is a stronger relationship between total water storage and surface inundation within the Amazon River in South America. Precipitation and surface inundation display stronger correlations within the Middle East compared to groundwater storage and surface inundation. It is clear that these single linear models are capable of
describing some surface inundation developments within specific regions, but not on a global scale.

There is a significant statistical improvement across the globe when including both groundwater storage and precipitation measurements in predicting surface inundation (Fig. 10c). Locations such as the Amazon, Mississippi and the Middle East have higher representation compared to the single linear models. The time-lag adjustment further improves our global correlations. Figures 11a and 11b display correlations with no time lag and 0 to 5 month time-lag corrections,
respectively. We can see visual improvements within the multi-linear regression's correlations east of the Andes and between the Sierra and the Rocky Mountains after the applied time lag correction.

Regression coefficient maps (Fig. 11c-f) have a color-axis between -1 to 1. Grey displays small values, and red represents large values. Regression coefficients for GPCP and GRACE from the non-time corrected model are shown in Fig. 11c and 11e while regression coefficients for GPCP and GRACE from the 0 to 5 months corrected model are displayed in Fig.
11d and 11f, respectively. White locations represent no data. The time lag correction moderates the extreme GPCP slopes around Northern Canada and Midwest North America. GRACE slopes around the Great Lakes and Australia also reflect this relationship.



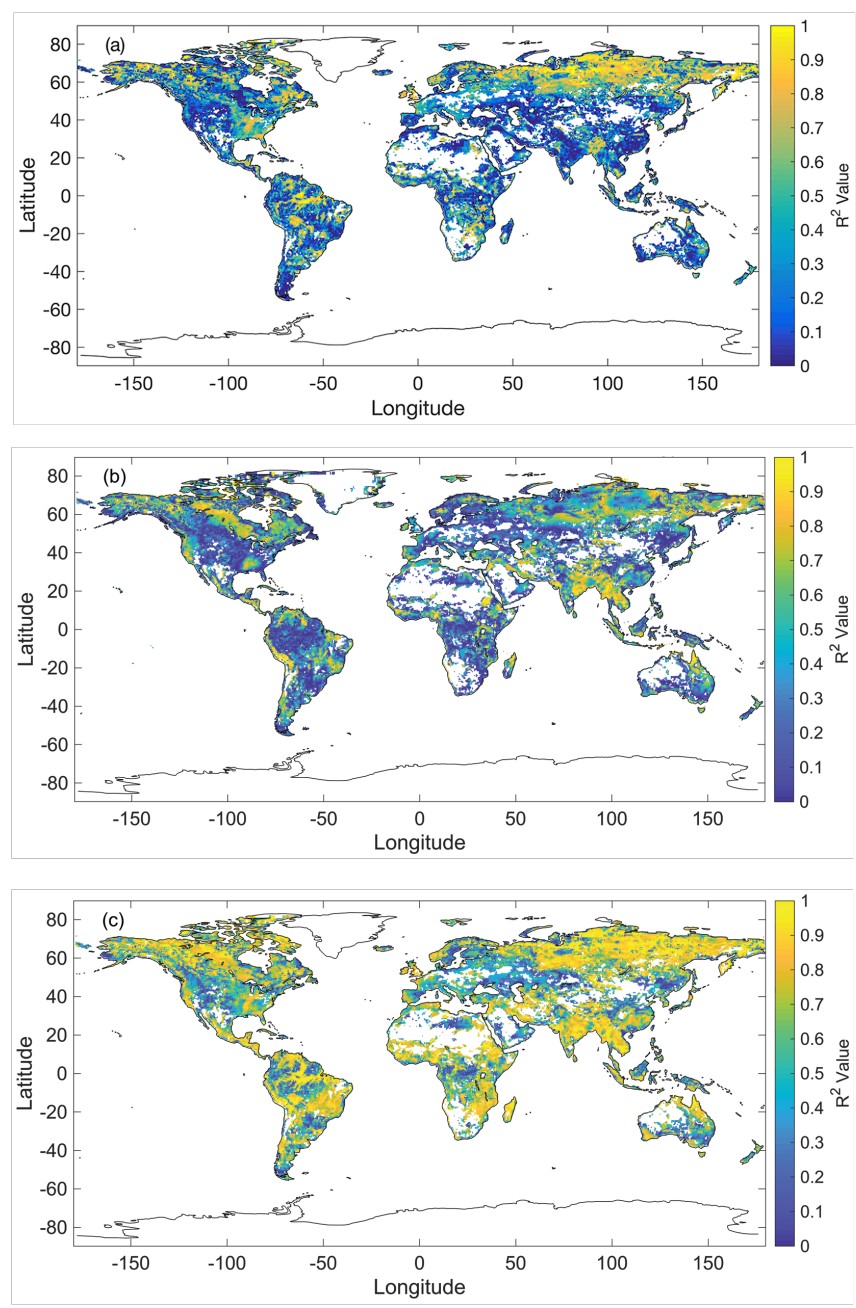

**Figure 10: Correlation maps for no time-lag corrected regression models a) Single linear regression between SWAMPS and GRACE. b) Single linear regression between SWAMPS and GPCP. c) Multi-linear regression between SWAMPS, GRACE, and GPCP.**



**Figure 11: a) Multi-linear regression correlations with no time correction. b) Multi-linear regression correlations with a time correction of 0 to 5 months. c) GPCP regression coefficients for the model in Fig. 10a. d) GPCP regression coefficients for the model in Fig. 10b. e) GRACE regression coefficients for the model in Fig. 10a. f) GRACE regression coefficients for the model in Fig. 10b.**




## 4 DISCUSSION

The surface water formation across the majority of locations within our study domain are controlled almost equally by groundwater storage and precipitation forcings. In our results, for the locations where precipitation has a substantial lag time, groundwater storage tends to have a smaller lag time. The converse is also true, and an inverse relationship follows for a

considerable GRACE lag and a slight GPCP lag. Sites such as the Amazon, Middle East, North America and parts of Asia reflect this pattern. Asia and the Middle East have larger lag times with groundwater storage compared to precipitation while the Amazon and North America have larger lag times with rainfall compared to groundwater storage.

By emphasizing the climatology, we created a model of inundation based on precipitation and storage that captures and predicts the average seasonal cycle. In areas that are profoundly affected by interannual variability, such as that during

ENSO events in locations such as Australia and Africa (Nicholson et al., 1997, Power et al., 1999, Ropelewski et al., 1987), our model under-predicts these infrequent anomalous fluxes. Heavy snow cover also creates detection issues within the SWAMPS surface water product. The effects of both snow and interannual variability may have influenced RMSE in these locations, and in general, the highest relative error occurs at high elevations and in locations that receive large amounts of snow, especially along the Rocky Mountains (Bales et al., 2006, Berghuijs et al., 2016, Yan et al., 2018). Rain-on-snow events

or rapid snowmelt could contribute to a rise in surface inundation without a relative increase in precipitation or groundwater storage. These types of situations are not considered or captured by our model.

No previous literature attempts to determine inundation developments with TWSA and precipitation measurements rather than just precipitation (Power et al., 1999, Prigent et al., 2007). However, there are studies on the watershed scale that have known control mechanisms. Papa et al. (2010) relate precipitation and river stage height to surface inundation extents

within the Amazon. They report precipitation to lead inundation with an influence of snow and glacier melt. We determined precipitation and storage are equally accountable for the inundation developments in the Amazon. Strong correlations between inundation, precipitation, and storage support our result. Papa et al. (2007) relate snowmelt and river discharge to surface inundation within the Ob basin. Maximum inundation is reported to occur between May and June with little to no lag between river discharge and maximum inundation. We report inundation in the Ob Basin as water storage driven and our reported lags

(maximum of one month) and modeled surface inundation climatology match their results. Temimi et al. (2005) predict flooding in the Mackenzie River Basin by relating river discharge to water surface fraction (WSF). The maximum flooding occurs during the spring when the snowpack melts and ice jams drive flooding. We report inundation developments to be controlled by both water storage and precipitation and the basin's modeled climatology reflects the same peak season.

Time lags between inundation and other variables have been well studied in hydrology (Hamilton et al., 2002, Power

et al., 1999, Prigent et al., 2007). Our reported precipitation time lags show similarity with those reported by Prigent et al. (2007) in the Amazon and South America. Instead of GRACE observations, Hamilton et al. (2002) correlated river stage observations to inundated areas. They report time lags between river stage and inundation for the Roraima and Pantanal floodplains in South America as 1 and 1.5-month lag. We report the lags for those areas to be two months. Their use of the





nearest river stage station and 0.25° cells of the Scanning Multi-channel Microwave Radiometer (SMMR) dataset compared
to the 0.5° cells of GRACE may account for this difference.

Our modeled inundation generally overpredicted locations with low surface inundation values. Areas along the Rocky
Mountains, northern parts of Russia and Asia all experienced overpredictions. Other studies on surface inundation have also
reported overestimations at locations with low inundation values (Prigent et al., 2007, Ticehurst et al., 2014). Issues such as
cloud coverage, fire scars, heavily snowed areas and large variation in topography could contribute to these over predictions.

**5 CONCLUSION**

This work relates global surface inundation developments to measurements of total water storage and precipitation using
NASA remote sensing observations. The novelty of this work is the combined application of the GRACE, GPCP and
SWAMPS data products to study and classify runoff generation mechanisms. We determine a majority of the global surface
inundation developments to be equally controlled by total water storage and precipitation. Our methods have the most
significant errors at locations with low values of inundation, which agrees with current literature. Remote sensing has provided
novel approaches to study general hydrology concepts on a global scale and holds much promise to further study phenomena
in areas with limited in situ data.

*Data Availability.* The data used in this work is publicly available. SWAMPS stable fractional surface inundation data can be
downloaded from Columbia University's International Research Institute for Climate and Society data library
(https://iridl.ldeo.columbia.edu/SOURCES/.NASA/.JPL/.wetlands/.dailyinundation/.swamps_v3p1/?Set-Language=en).
GPCP monthly average precipitation data is provided by NOAA/OAR/ESRL PSD, Boulder, Colorado, USA, at
https://www.esrl.noaa.gov/psd/data/gridded/data.gpcp.html. GRACE Mascon data are available at http://grace.jpl.nasa.gov,
supported by the NASA MEaSUREs Program.

*Author contributions.* J.T. Reager and S. R. Lopez conceptualized, funded, and supervised this work; J. T. D. Lucey conducted
the primary investigation, visualization, and formal analysis for this work. J. T. D. Lucey prepared the publication with
contribution from both co-authors.

*Competing Interest.* The authors declare that they have no conflict of interest.

*Acknowledgments.* A portion of this work was conducted at the Jet Propulsion Laboratory, California Institute of Technology,
under contract with NASA. This work was funded by the NASA DIRECT-STEM Center (NASA Award Number
NNX15AQ06A), NSF LSAMP program at California State University, Los Angeles, and the NASA GRACE Science Team.



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





| Model | Lag Correction | R² No QC | RMSE | Coverage No QC [%] | R²/ Coverage [-] | R² QC ≥ 50 | RMSE | Coverage QC ≥ 50 [%] | R²/ Coverage [-] |
|---|---|---|---|---|---|---|---|---|---|
| GPCP+GRACE | None | 0.760 | 3.64 | 97.25 | 0.78 | 0.896 | 1.94 | 77.71 | 1.15 |
| GPCP+GRACE | 0 to 5 | 0.732 | 3.83 | 97.12 | 0.75 | 0.900 | 1.89 | 77.58 | 1.16 |
| GPCP+GRACE | 0 to 11 | 0.730 | 3.85 | 97.12 | 0.75 | 0.901 | 1.89 | 77.58 | 1.16 |
| GPCP | None | 0.911 | 3.37 | 97.64 | 0.93 | 0.974 | 1.46 | 78.10 | 1.25 |
| GRACE | None | 0.788 | 3.42 | 97.25 | 0.85 | 0.899 | 1.90 | 77.71 | 1.16 |
| GPCP | 0 to 5 | 0.887 | 3.79 | 97.64 | 0.91 | 0.968 | 1.64 | 78.10 | 1.24 |
| GRACE | 0 to 5 | 0.692 | 4.11 | 97.12 | 0.71 | 0.856 | 2.28 | 77.58 | 1.10 |
| GPCP | 0 to 11 | 0.887 | 3.79 | 97.64 | 0.91 | 0.968 | 1.64 | 78.10 | 1.24 |
| GRACE | 0 to 11 | 0.692 | 4.12 | 97.12 | 0.72 | 0.856 | 2.28 | 77.58 | 1.10 |

**Table 1: Model validation results; QC = Quality control, RMSE = Root mean squared error**



| Site | Amazon | | Mackenzie | | Mississippi | | Ob | |
|---|---|---|---|---|---|---|---|---|
| | Longitude | Latitude | Longitude | Latitude | Longitude | Latitude | Longitude | Latitude |
| Green | -52.25 | -1.25 | -119.25 | 61.25 | -89.75 | 32.75 | 71.25 | 60.75 |
| Blue | -65.25 | -2.25 | -125.75 | 63.75 | -88.75 | 37.25 | 80.75 | 56.25 |
| Red | -56.25 | -2.25 | -131.25 | 66.25 | -89.75 | 35.35 | 76.25 | 59.25 |
| Boundary | -76.25 to -52.25 | -9.75 to 3.25 | -134.25 to -112.75 | 56.75 to 67.75 | -91.25 to -87.75 | 31.25 to 38.75 | 69.25 to 81.75 | 55.75 to 65.25 |

**Table 2: Coordinates for basin sites and the boundaries for cells included in the scatterplots**




| Site | Amazon | | Mackenzie | | Mississippi | | Ob | |
|------|--------|------|-----------|------|-------------|------|------|------|
| | R² | RMSE | R² | RMSE | R² | RMSE | R² | RMSE |
| Green | 0.817 | 1.275 | 0.967 | 0.290 | 0.776 | 0.082 | 0.868 | 0.947 |
| Blue | 0.889 | 0.455 | 0.955 | 0.009 | 0.855 | 0.389 | 0.886 | 0.544 |
| Red | 0.916 | 1.356 | 0.994 | 0.148 | 0.855 | 0.466 | 0.909 | 0.265 |

**Table 3: Basin climatology correlation statistics**