# Peer review of "Global partitioning of runoff generation mechanisms using remote sensing data"

_Hydrology and Earth System Sciences, 2019_

## Referee Comment (RC1) · Anonymous Referee #1 · 25 Jul 2019

Review of 'Global partitioning of runoff generation mechanisms using remote sensing data' by Lucey, Reager and Lopez, hess-2019-292

Lucey et al. provide a method to assess contributions of precipitation and water storage to runoff generation at the global scale. The scientific significance is their use of global satellite data to study the two contributions at global scale, the paper therefore fits within the scope of HESS. The methods themselves seem valid, but the paper needs revisions mainly for clarification purposes. For instance, the abstract, intro ('our goal is') and conclusion can be more in line, and I do not understand why two time-lag thresholds are applied. Detailed comments are given below. I would therefore say the scientific significance is good-excellent, while the scientific and presentation quality is fair.

P1 l16-18: This last line of the abstract is not a conclusion that I got from reading the paper. Now it seems that this is one of your main points, whereas in the text you find that most areas have mixed time lags and are driven by both precipitation and water storage.

P2 l56-58. Here you clearly state 3 research questions. These are not reflected well in the abstract. Also, I feel that the characterization of general behavior is limited.

P5 l87: you mention the common period amongst the 3 satellite products is April 2002-October 2015, but the previous paragraph mention that all satellite products are available until 2017 or present. So why October 2015?

P5 l 96: is each dataset's climatology based on the 2002-2015 period? Then I would not call it a climatology as it only spans 13 years, rather 'the 2002-2015 average'.

P5 l105-107: I don't understand the concept of the two time-lag thresholds. Do you check the cross-correlations for up to 5 months lag as well as up to 11? Also, if you are using a climatology (i.e. 12 months), going up to 11 months lag makes little sense to me to start with.

P5 l 114: please explain what coverage means. E.g. grid cells covered by swamps? (I'm also not very familiar with GIS so the term polygon is also unfamiliar to me).

P6 eq. 4-6: abbreviation LTA is not explained here. Also does 'slope' refer to m1, m2 in Equation 3? I thought those were determined using the climatologies, but then determining the standardized values (Eq 5) would not make sense, hence I'm confused at the methods here. Or are these equations only used for the four highly studied basins, i.e. Eq 5 shows the standardized value determined by the gridcells in the basins? Also, Eq 6 could be re-written to make it clearer that the control variable is determined by the slopes, e.g. slopegpcp - slopegrace.

P8 l150: "we no longer consider all 0-11 month models", yet it is shown in Fig 9.

P8 Fig4: just a small suggestion to make the figures easier to read, at least in my

opinion. Maybe place a small text within the figures, rather than having the reader disentangle which figure represents which dataset and which lag threshold (also goes for other figures).

P9 l 164: "we can see" ... "we know". Actually I can't see or know because so far I've only seen the results of the multi-linear regression (Fig 5) and not the single regression models. The comparison for single and multi-linear regression only shows up in Fig 10 (without time lags). So I would rephrase or re-order, as you mention in this paragraph that a 'multi-linear regression model with a time lag correction between 0 and 5 months is the most rigorous for further analysis', so I was a bit surprised to see Fig 10 discussed later on.

P10 l 173: I assume you mean August 2007 rather than specifically August 15th 2007.

P10 l 177-179: I got a bit confused which dataset has which limitations in which locations. SWAMPS data has limitations over desert and mountainous areas shown in Fig 1, but modeled SWAMPS (maybe better to call it modeled inundation rather than SWAMPS?) has limitations in areas with snow and ice or seasonal monsoon areas, so that is related to limitations in either GPCP or GRACE?

P11 l198-199: related to the above, is the inadequate data related to GPCP or GRACE?

P12 Fig 7: if there are no available measurements in winter, then the scatter plots reflect only the summer / fall months?

P13 l210-216: refer back to Eq 4-6 to help the reader remember how you determined your error.

P14 l224: "white areas represent no values" this is repeatedly mentioned in the text but not in the captions. It would be OK to mention this clearly once (white areas are masked using SWAMPS quality map).

P15 Fig 9: refer to Eq 6

P16 l241: bracket goes before 'comparing' instead of 'Fig'.

P16 l252-257: Regression coefficient refers to Eq 3 (m1, m2, called slope elsewhere)? Furthermore, you use a scale from -1 to 1 whereas in lines 99-107 you mention that negative regression coefficients should be impossible, and therefore you introduce time lags. So why are there still negative values in Figure 11? Should results for those grid cells not be trusted? The color scale is also a bit misleading, as grey values (towards -1) do not reflect small values (around 0), but the orange colors do.

Overall: I felt it was a bit confusing that the terms GRACE / TWSA / water storage, GPCP / precipitation, SWAMPS / runoff generation / inundation are used interchangeably.

---

## Author Comment (AC1) · 7 Sep 2019

Author responses are started with a "-" and placed below each comment.

P1 l16-18: This last line of the abstract is not a conclusion that I got from reading the paper. Now it seems that this is one of your main points, whereas in the text you find that most areas have mixed time lags and are driven by both precipitation and water storage.

- Line 16 – 18 will be changed to "Precipitation and total water storage equally control the majority of surface inundation developments across the globe. The model tends to underestimate and overestimate at locations with high interannual variability and with low inundation measurements, respectively" to properly state our conclusions.

[Figure]

P2 l56-58. Here you clearly state 3 research questions. These are not reflected well in the abstract. Also, I feel that the characterization of general behavior is limited.

Is there a typo on line 56? Is "too" supposed to be "to" for "We conduct such a study here too:. . .?"

- Lines 9 – 11 in the abstract touch on these research questions. I will update the sentence to "In this study, we examine the covariance of global satellite-based surface water inundation observations with two remotely sensed hydrological variables, precipitation, and terrestrial water storage, to better understand how apparent runoff generation responds to these two dominant forcing mechanisms in different regions of the world" in the final manuscript as it includes reference to our regional analysis.

P5 l87: you mention the common period amongst the 3 satellite products is April 2002-October 2015, but the previous paragraph mention that all satellite products are available until 2017 or present. So why October 2015?

- When I first accessed the data, GPCP was only available until Oct. 2015. While I was writing the manuscript, the extent of the data had been updated and we did not extend the analysis period.

P5 l 96: is each dataset's climatology based on the 2002-2015 period? Then I would not call it a climatology as it only spans 13 years, rather 'the 2002-2015 average'.

- The climatology of a region is defined as weather conditions averaged over a period of time. We feel this is a sufficient time frame to use for a climatology.

P5 l105-107: I don't understand the concept of the two time-lag thresholds. Do you check the cross-correlations for up to 5 months lag as well as up to 11? Also, if you are using a climatology (i.e. 12 months), going up to 11 months lag makes little sense to me to start with.

- Correct, we check the cross-correlations for up to 5 months lag as well as up to 11. We check up to 11 months to ensure there are no erroneous lags and no significant

statistical improvement in the results.

P5 l 114: please explain what coverage means. E.g. grid cells covered by SWAMPS? (I'm also not very familiar with GIS so the term polygon is also unfamiliar to me).

- Lines 114 – 116 will be changed to, "Pearson's R2, the root mean squared error (RMSE), and a ratio between R2 and coverage were used to determine each model's strength. Coverage is considered the number of SWAMPS grid cells with numerical values within the global coastline; for example, the analysis excluded Antarctica and Greenland because there is no SWAMPS data for these regions." to clarify our use of coverage.

P6 eq. 4-6: abbreviation LTA is not explained here. Also does 'slope' refer to m1, m2 in Equation 3? I thought those were determined using the climatologies, but then determining the standardized values (Eq 5) would not make sense, hence I'm confused at the methods here. Or are these equations only used for the four highly studied basins, i.e. Eq 5 shows the standardized value determined by the gridcells in the basins? Also, Eq 6 could be re-written to make it clearer that the control variable is determined by the slopes, e.g. slopegpcp - slopegrace.

- The "measured SWAMPS signal" (Line 129) will be changed to "measured SWAMPS long term average (LTA)". "Slope" in general refers to the m variables and m1 and m2 are specifically used to determine the control variable. Slopes at each grid cell are determined through the regressions which utilize our developed climatologies. We standardized the calculated slopes to determine the control variable. Since GRACE and GPCP function on different ranges and units, we can expect the slopes to reflect the spread and magnitude of their dataset. We remove the slopes' spread and magnitude by subtracting and dividing by the average and standard deviation of all the calculated slopes per variable. Therefore, when we use Equation 6 our inputs are fairly compared. Equation 6 will be rewritten as: Control Variable= |GPCP Slope|-|GRACE Slope|

P8 l150: "we no longer consider all 0-11 month models", yet it is shown in Fig 9.

- 0 – 11 month model results are provided to provide a thorough analysis. We felt readers would like to see the insignificant differences between the 0 – 5 and 0 – 11 month models for themselves.

P8 Fig4: just a small suggestion to make the figures easier to read, at least in my opinion. Maybe place a small text within the figures, rather than having the reader disentangle which figure represents which dataset and which lag threshold (also goes for other figures).

- The authors feel the figure captions sufficiently state which dataset and lag threshold are applied per (sub)figure.

P9 l 164: "we can see" . . . "we know". Actually I can't see or know because so far I've only seen the results of the multi-linear regression (Fig 5) and not the single regression models. The comparison for single and multi-linear regression only shows up in Fig 10 (without time lags). So I would rephrase or re-order, as you mention in this paragraph that a 'multi-linear regression model with a time lag correction between 0 and 5 months is the most rigorous for further analysis', so I was a bit surprised to see Fig 10 discussed later on.

- I am referencing the validation statistics provided in Table 1 in Line 164. I will add "(Table 1)" after "there isn't much improvement" to clarify this statement is a result of the validation results in Table 1.

P10 l 173: I assume you mean August 2007 rather than specifically August 15th 2007.

- Yes, thank you. This will be changed to "August 2007" in the final revision to prevent confusion.

P10 l 177-179: I got a bit confused which dataset has which limitations in which locations. SWAMPS data has limitations over desert and mountainous areas shown in Fig 1, but modeled SWAMPS (maybe better to call it modeled inundation rather than SWAMPS?) has limitations in areas with snow and ice or seasonal monsoon areas, so

that is related to limitations in either GPCP or GRACE?

- The limitations of SWAMPS are reflected in modeled SWAMPS. Areas heavy in snow and ice often report missing values or inconsistent measurements in SWAMPS (example in Figure 7e). We attribute the underestimations in areas affected by large interannual variability to the limited shared data period between the datasets and to utilizing climatologies which would reflect the average behavior.

P11 l198-199: related to the above, is the inadequate data related to GPCP or GRACE?

- Areas heavy in snow and ice often report missing values or inconsistent measurements in SWAMPS (example in Figure 7e).

P12 Fig 7: if there are no available measurements in winter, then the scatter plots reflect only the summer / fall months?

- These validation plots reflect the modeled versus measured inundation per month within the domain (163 months between April 2002 to October 2015). It is possible there were no measurements at a location for all winters, no measurements for only a single winter month (NaNs for Dec., values for Jan.), or no measurements during a specific year (NaNs for Winter 2007, values for Winter 2008). With this understanding, we expect the scatter plots in snow/ice dominated regions to reflect some winter trends, but not as thorough as locations that are not limited by snow/ice.

P13 l210-216: refer back to Eq 4-6 to help the reader remember how you determined your error.

- I will add text that references the equations in the final revision.

P14 l224: "white areas represent no values" this is repeatedly mentioned in the text but not in the captions. It would be OK to mention this clearly once (white areas are masked using SWAMPS quality map).

- White areas represent low values and/or no value depending on the figure and is stated whenever referenced for the reader. I would prefer to leave the text referencing the white as is to prevent possible confusion.

P16 l241: bracket goes before 'comparing' instead of 'Fig'.

- Thank you for catching this. We will remove "comparing" and the following comma in the final revision.

P16 l252-257: Regression coefficient refers to Eq 3 (m1, m2, called slope elsewhere)? Furthermore, you use a scale from -1 to 1 whereas in lines 99-107 you mention that negative regression coefficients should be impossible, and therefore you introduce time lags. So why are there still negative values in Figure 11? Should results for those grid cells not be trusted? The color scale is also a bit misleading, as grey values (towards -1) do not reflect small values (around 0), but the orange colors do.

- We hypothesized it should be generally impossible to have negative slopes because we wouldn't expect GRACE or GPCP to be inversely related to SWAMPS. However, we see this occur within the data before a time lag correction (Fig 11c and 11f) and after the correction (Fig. 11d and 11f). Some of the negative values are attributed to time lag and are reduced through the applied corrections. Other negative values tend to occur in regions with known limitations (areas of high elevation, snow dominated, and poor SWAMPS reliability) and are discussed in the manuscript. My apologies on the wording. The phrase will be changed to "Grey displays negative values" in the final submission.

- Figure 11c and 11e reflect slopes for the model in Figure 11a (multi-linear regression with no time lag). Figure 11d and 11f reflect slopes for the model in Figure 11b (multi-linear regression a time correction of 0 to 5 months). The caption was mislabeled and will be corrected.

Overall: I felt it was a bit confusing that the terms GRACE / TWSA / water storage,

[Figure]

GPCP / precipitation, SWAMPS / runoff generation / inundation are used interchange-ably.

- These satellites and their respective measurements are core terminology within the literature.

---

## Referee Comment (RC2) · Anonymous Referee #2 · 5 Nov 2019

This manuscript assessed a linear regression model for predicting surface inundation (SWAMPS) using two predictors, gridded precipitation product (GPCP) and GRACE TWS anomalies (TWSA). While the problem may be interesting, the hypothesis and approach taken were too naive or even erroneous. - TWS is highly correlated to precipitation (with potential lag adjustments) in most areas (see Humphrey et al., 2016, Figure 8a). From linear regression theory, any time you have collinearity, the results may look weird. - The authors' hypothesis is that surface water inundation is linearly correlated to precipitation amount. This may not be true from topography perspective. As a case in example, heavy storm in mountainous areas will result in different inundation extent than that in plain areas. - SWAMPS is daily data and surface inundation is often more meaningful at daily or even subdaily scales. At the monthly scale, SWAMPS

basically shows surface water bodies that can be well delineated from Landsat data. Why do we need your model in the first place? - I looked at the GPCP description, which says "The Version 2.3 monthly product covers the period January 1979 to the present, with a delay of two to three months for data reception and processing." Similarly, GRACE monthly data product has several months of data processing latency. So the significance of this linear regression approach (requiring GRACE and GPCP) for flood inundation is limited. - The scale 0.25deg is too coarse for water managers who are interested in inundation.

Reference: Humphrey, V., Gudmundsson, L., & Seneviratne, S. I. (2016). Assessing global water storage variability from GRACE: Trends, seasonal cycle, subseasonal anomalies and extremes. Surveys in Geophysics, 37(2), 357-395.

---

## Author Comment (AC2) · 17 Dec 2019

We thank the reviewer for their comments and wish to address each of them in detail. The reviewer has raised some valid concerns, from the perspective of improving the articulation of the motivation and approach in our work. However, there is some fundamental misunderstanding in this review that needs to be addressed in detail.

We see the reviewer's comments are distributed across three major contextual points. First, the reviewer appears to assume that the motivation of this study was to develop a sort of operational predictive tool, for which data latency would be a critical issue and for which the spatial scale of the study may be limiting. In fact, that assumption is erroneous; this study was never focused on addressing an operational need at all. It is

instead a scientific study of processes critical to better understanding global and large scale hydrology. The purpose of our work was to explore and characterize surface inundation developments with precipitation and water storage for the first time using NASA remote sensing data products. To our knowledge, this was a first attempt at considering two contributors in surface inundation generation and attempting to understand "which process dominates inundation, and where." These insights could be useful to hydrologists and global land surface modelers, but are not intended to be used for operational forecasting.

We would like to take specific note of the reviewer's comment, "Why do we need your model in the first place?", which we honestly find a bit shortsighted. To clarify, we are not proposing an operational "model" here at all; we are conducting a study of processes and their spatial distribution globally. As scientists, we perform studies to better understand the mechanisms and processes that cause the phenomena we observe. Then we assemble these studies into a manuscript and publish it to advance the community knowledge of that phenomena. That is "why we need" this study, and why we need science in general. The question of "why do we need your model in the first place" presents a bit of a ridiculous perspective.

Towards the exploration of observed surface water generation, we apply a regression model framework to better understand mechanisms, but the model itself is not a product or an outcome. It is simply a tool to address the mapping of dominant processes. This difference in motivation is important in understanding the paper we believe and it seems the reviewer has missed that point with this question.

We have modified the manuscript to make this point more clear, so that there is no confusion between a scientific process study and an operational development study. Though we had never mentioned an operational motivation for this work, we have now removed all text that may have implied an operational need, and changed the lines listed below to now read as follows:

- From "prediction" to "estimation" (Line 15).

- "We approach these goals through the application of a simple linear regression model of inundation based on remote sensing observations" (Line 58).

- From "predict surface inundation" to "represent surface inundation" (Line 89).

- "To further capture the long-term variability across the globe, we utilized each dataset's climatology" (Line 95).

- From "developed model" to "regressions" (Line 103).

- "With the final model, historical GRACE and GPCP measurements are used to estimate surface inundation (referred to as modeled surface inundation)" (Lines 120).

- From "predicts" to "estimates" (Line 279).

Second, it seems that the reviewer was concerned with the coarse scale of the study, but also simultaneously concerned that topographic heterogeneity will drive inundation patterns at fine scales. These comments can be read as inconsistent but hopefully, we can clarify our approach. For this study, we imagine a global land surface model, typically run at 1 degree globally (or at best, 0.25 degrees globally), for which topographic processes are represented empirically, and in which surface water formation follows Beven and Kirkby's 'topmodel' formulation. In this, topography and topographic heterogeneity are represented statistically, and there are truly still aggregated (or "lumped') runoff generation processes that occur at coarse resolution. At those scales, the topography is never explicitly represented, but instead, is represented implicitly as a grid-cell level characteristic that can influence lumped runoff generation. Here we have taken the same conceptual approach, for which we examine the aggregated runoff generation across the entire 0.25-degree grid cell, and those results can be associated with topographic information but without an explicit representation of topography in the regression. This is a simple and valid approach that is observation-focused, to later diagnose processes and mechanisms statistically.

To clarify this fact for readers of the study, we have added text to this effect, between lines 54 and 55 in the manuscript.

Third, it seems that the reviewer is not convinced of the orthogonality of precipitation and terrestrial water storage anomaly time series. In fact, as the reviewer has highlighted and as explained in Humphrey et al., 2016, and many, many excellent papers before that one, there is approximately a 3-month time lag between precipitation (a flux), and storage (a state), on average globally. This time lag between the rate and the state does, in fact, create orthogonality between the two-time series, similar to the orthogonality between a sine and a cosine wave. Leveraging this orthogonality is what allows us to apply a multiple regression and disaggregate the effects of these two processes. That is the entire premise of the approach, so we empathize that having misunderstood this fact, the reviewer would be confused by our methodology.

To make this point more clearly, we have added text in the method section on the orthogonality of precipitation and storage time series (at the end of line 89):

"Precipitation and water storage long-term anomalies, a component of the total signal, are known to be globally correlated with a known lag (Humphrey et al., 2016). We utilize full signal in the regressions to ensure levels of orthogonality between precipitation and water storage that avoid collinearity."